# Plk1 regulates spindle orientation by phosphorylating NuMA in human cells

Shrividya Sana*, Riya Keshri*, Ashwathi Rajeevan, Sukriti Kapoor, Sachin Kotak

**Proper orientation of the mitotic spindle defines the correct division plane and is essential for accurate cell division and development. In metazoans, an evolutionarily conserved complex comprising of NuMA/LGN/Gαi regulates proper orientation of the mitotic spindle by orchestrating cortical dynein levels during metaphase. However, the molecular mechanisms that modulate the spatiotemporal dynamics of this complex during mitosis remain elusive. Here, we report that acute inactivation of Polo-like kinase 1 (Plk1) during metaphase enriches cortical levels of dynein/NuMA/LGN and thus influences spindle orientation. We establish that this impact of Plk1 on cortical levels of dynein/NuMA/LGN is through NuMA, but not via dynein/LGN. Moreover, we reveal that Plk1 inhibition alters the dynamic behavior of NuMA at the cell cortex. We further show that Plk1 directly interacts and phosphorylates NuMA. Notably, NuMA-phosphorylation by Plk1 impacts its cortical localization, and this is needed for precise spindle orientation during metaphase. Overall, our finding connects spindle-pole pool of Plk1 with cortical NuMA and answers a long-standing puzzle about how spindle-pole Plk1 gradient dictates proper spindle orientation for error-free mitosis.**

## Introduction

The precise orientation of the mitotic spindle determines the correct placement of the cleavage furrow and thus maintains the relative sizes and spatial organization of the daughter cells. Proper orientation of the mitotic spindle further ensures that the cell fate determinants are accurately segregated in the resulting daughter cells during asymmetric cell division, including in stem cells. In metazoans, spindle orientation is regulated by an evolutionarily conserved ternary complex consisting of a large coiled-coil protein, a GoLoCo domain–containing protein, and heterotrimeric G protein α subunit (NuMA/LGN/Gαi in humans; reviewed in Siller & Doe [2009], di Pietro et al [2016], Seldin & Macara [2017], Bergstralh et al [2017]). This complex serves to anchor the minus-end–directed motor protein complex dynein (hereafter referred to as dynein) at the cell cortex beneath the plasma membrane (reviewed in Kotak & Gönczy [2013]). Such cortically anchored dynein is thought to regulate spindle orientation by walking over the dynamic astral microtubules and thus exerting the pulling forces on the astral microtubules and therefore on the spindle apparatus (Nguyen-Ngoc et al, 2007; Kotak et al, 2012; Laan et al, 2012).

NuMA acts as an essential adaptor molecule for anchoring cortical dynein both in metaphase (Du & Macara, 2004; Woodard et al, 2010; Kiyomitsu & Cheeseman, 2012; Kotak et al, 2012) and during anaphase (Kiyomitsu & Cheeseman, 2013; Kotak et al, 2013; Seldin et al, 2013; Zheng et al, 2014). Besides its role in orchestrating spindle orientation, NuMA is required for the proper assembly of the mitotic spindle (Compton et al, 1992; Yang & Snyder, 1992; Merdes et al, 1996). In mitosis, NuMA interacts with dynein through its N-terminus region and associates with LGN and microtubules through its C-terminus (Merdes et al, 1996; Du et al, 2002; Kotak et al, 2012, 2014; Gallini et al, 2016; Hueschen et al, 2017). Because NuMA acts as an essential adaptor molecule for dynein during mitosis, and this property of NuMA helps in coordinating several mitotic events; its localization must be tightly regulated in a spatiotemporal manner. Interestingly, NuMA cortical levels are dynamically modulated by several vital mitotic kinases. For instance, NuMA is shown to be directly phosphorylated by Cdk1/cyclinB (Kotak et al, 2013), and this phosphorylation negatively impacts cortical accumulation of NuMA and thus dynein during metaphase (Kiyomitsu & Cheeseman, 2013; Kotak et al, 2013; Seldin et al, 2013; Zheng et al, 2014). Moreover, Aurora A was recently identified as a potential kinase that affects spindle orientation by phosphorylating and thus modulating the levels of cortical NuMA (Gallini et al, 2016; Kotak et al, 2016).

Polo-like kinase 1 (Plk1) is an essential serine–threonine kinase that was initially identified in flies (Sunkel & Glover, 1988) and it is indispensable for several mitotic events in all the organisms studied to date (reviewed in Archambault & Glover [2009], Bruinsma et al [2012]). Plk1 is characterized by Polo-box domain (PBD) that acts as a phosphopeptide-binding site and targets Plk1 to several subcellular locations (reviewed in van de Weerdt & Medema [2006], Archambault & Glover [2009]). In mammals, Plk1 regulates a considerable number of mitotic processes including centrosome

Department of Microbiology and Cell Biology, Indian Institute of Science (IISc), Bangalore, India

Correspondence: sachinkotak@iisc.ac.in
*Shrividya Sana and Riya Keshri contributed equally to this work.

maturation, bipolar spindle assembly, attachment of microtubules to the kinetochore, and cytokinesis (Barr et al, 2004; Peters et al, 2006; Lenart et al, 2007; Petronczki et al, 2007; Burkard et al, 2009). In the past few years, a large number of studies have linked Plk1 function with proper spindle orientation. For instance, Plk1 is shown to regulate an actin-associated protein MISP that influence spindle orientation by affecting astral microtubules (Zhu et al, 2013), and more recently, several genes such as WDR62/MCPH2, NDR1, and HMMR have been shown to be a part of Plk1-mediated spindle orientation pathway (Connell et al, 2017; Miyamoto et al, 2017; Yan et al, 2015). Importantly, in the context of this study, the spindle-pole pool of Plk1 was implicated in negatively regulating cortical dynein localization during metaphase (Kiyomitsu & Cheeseman, 2012; Collins et al, 2012). Moreover, it is suggested that the spindle-pole–localized Plk1 in prometaphase acts as a molecular ruler that negatively controls the cortical levels of LGN and thus dynein by sensing the distance from the spindle-pole to the cell cortex (Tame et al, 2016). Intriguingly, it is not clear if Plk1 also impacts cortical localization of the components of the ternary complex during anaphase, and more importantly, what is the direct target of Plk1 in modulating cortical dynein levels for proper spindle behavior remains unknown.

In this study, by utilizing a small molecule inhibitor for Plk1 (McInnes et al, 2006; Peters et al, 2006; Steegmaier et al, 2007), we report that acute Plk1 inhibition in metaphase and in anaphase robustly enriches NuMA and dynein at the cell cortex in human cells. Cortical NuMA enrichment upon Plk1 inhibition appears to stem from its decrease in the mobility at the cell cortex, and this leads to spindle orientation defects in metaphase. Furthermore, we uncover that Plk1 directly interacts with and phosphorylates cortical dynein adaptor NuMA. Importantly, we identified Plk1 phosphorylation sites at the C-terminus of NuMA, which when mutated to alanine can impact cortical NuMA levels by modulating its turnover and thus lead to spindle orientation defects in human cells. In summary, our study identifies NuMA as a primary substrate of Plk1 and answers how the spindle-pole pool of Plk1 may modulate spindle orientation for flawless completion of mitosis.

# Results and Discussion

### Acute inactivation of Plk1 enriches cortical levels of NuMA/dynein and LGN during metaphase

Hourly long Plk1 inactivation using BI 2536 blocks bipolar spindle assembly during mitosis (Steegmaier et al, 2007). Therefore, to study the impact of acute Plk1 inactivation on the cortical levels of NuMA/LGN/Gαi$_{1-3}$ and dynein during metaphase, we adopted a multi-drug strategy that has earlier been used to study the impact of Plk1 on cytokinesis (Petronczki et al, 2007) (Fig S1A). In brief, post–thymidine-treated HeLa cells were synchronized in pro-metaphase using spindle poison Nocodazole and afterwards blocked in metaphase using proteasome inhibitor MG132. Subsequently, these cells were released from the MG132 block and were treated with either DMSO (control) or Plk1 inhibitor (BI 2536; 100 nM) (see Fig S1A figure legend for details). This treatment resulted in synchronous entry into anaphase for the majority of the cells, for both the control and BI

2536–treated cells (Fig S1B and C), suggesting that Plk1 inhibition in our setting is not provoking spindle checkpoint activation during mitosis. 12 h of BI 2536 treatment post–MG132 release resulted in substantial enrichment in the binucleated cells because of cytokinesis failure, indicating the potency of the Plk1 inhibitor in our experimental setup (Fig S1D–F) (Petronczki et al, 2007). Next, we analyzed the cortical localization of NuMA, dynein-associated dynactin component p150$^{Glued}$, LGN, and Gαi$_3$ at 20, 40, and 60 min post BI 2536 exposure (Fig S1G–L and data not shown). Cortical levels of NuMA, p150$^{Glued}$, and LGN were markedly increased in the polar regions of the cell cortex upon incubation with BI 2436 for 40 min, in contrast to the Gαi$_3$ that remained unaltered (compare Fig S1H with S1G, S1L with S1K and S1I with S1J). Similar results were obtained by acutely treating non-synchronized cells with BI 2536 (300 nM) for 30 min (compare Fig 1B with 1A), and this regimen is used thereafter for all the experiments. Likewise, acute inactivation of Plk1 using BI 2536 in non-transformed hTERT1-RPE1 cells also led to the significant enrichment of cortical NuMA during metaphase suggesting that the increase in the cortical NuMA levels upon Plk1 inhibition is not restricted to HeLa cells (Fig S2A and compare Fig S2D with S2B). Cortical NuMA enrichment upon BI 2536 treatment appears to be Plk1 specific as partial depletion of Plk1 using siRNAs targeting 3′ UTR of Plk1 also resulted in enhancement of cortical NuMA localization during mitosis (Fig S2F–M). These data suggest that Plk1 negatively controls NuMA/LGN and dynein localization during metaphase.

### Dynein is dispensable for cortical NuMA localization enrichment upon Plk1 inhibition during mitosis

Previously, cortical dynein levels were reported to be affected by Plk1 (Kiyomitsu & Cheeseman, 2012). This prompted us to investigate the dependency of cortical NuMA on dynein in metaphase cells treated with Plk1 inhibitor BI 2536. As mentioned above, acute inactivation of Plk1 leads to strong enrichment of endogenous NuMA/p150$^{Glued}$ (compare Fig 1B with 1A) or ectopically expressed GFP-NuMA during metaphase (compare Fig 1H with 1F). Notably, Plk1 inactivation in cells treated with dynein heavy chain (DHC1) siRNAs resulted in robust enrichment of cortical NuMA similar to the control cells (compare Fig 1D with 1C). To corroborate this finding, we utilized cells expressing the C-terminus of NuMA [GFP-NuMA$_{(1,411–2,115)}$] that localize to the cell cortex analogous to the full-length NuMA, but unable to interact with dynein (Fig 1E) (Kotak et al, 2012, 2013). We exposed GFP-NuMA$_{(1,411–2,115)}$ expressing cells to BI 2536. Similar to the results obtained for the endogenous NuMA and GFP-NuMA, acute inactivation of Plk1 resulted in substantial enrichment of GFP-NuMA$_{(1,411–2,115)}$ in metaphase (compare Fig 1L with 1J). Altogether, these data suggest that cortical NuMA enrichment upon Plk1 inactivation is dynein-independent.

### LGN is required for Plk1-dependent cortical NuMA enrichment in metaphase but not during anaphase

Next, we assess the consequence of LGN depletion on metaphase cortical NuMA in cells simultaneously inactivated for Plk1. LGN is required for cortical NuMA/p150$^{Glued}$ localization during metaphase (Du & Macara, 2004; Woodard et al, 2010; Kiyomitsu & Cheeseman, 2012; Kotak et al, 2012). Therefore, as expected, RNAi-mediated

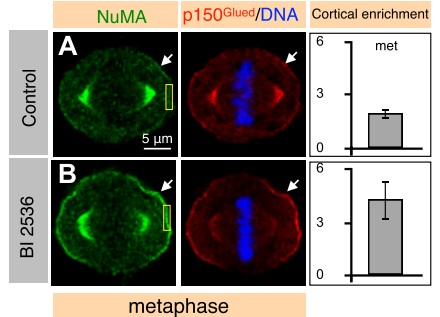

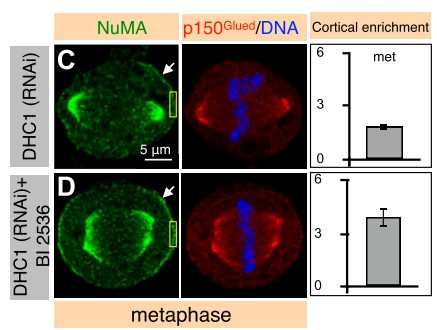

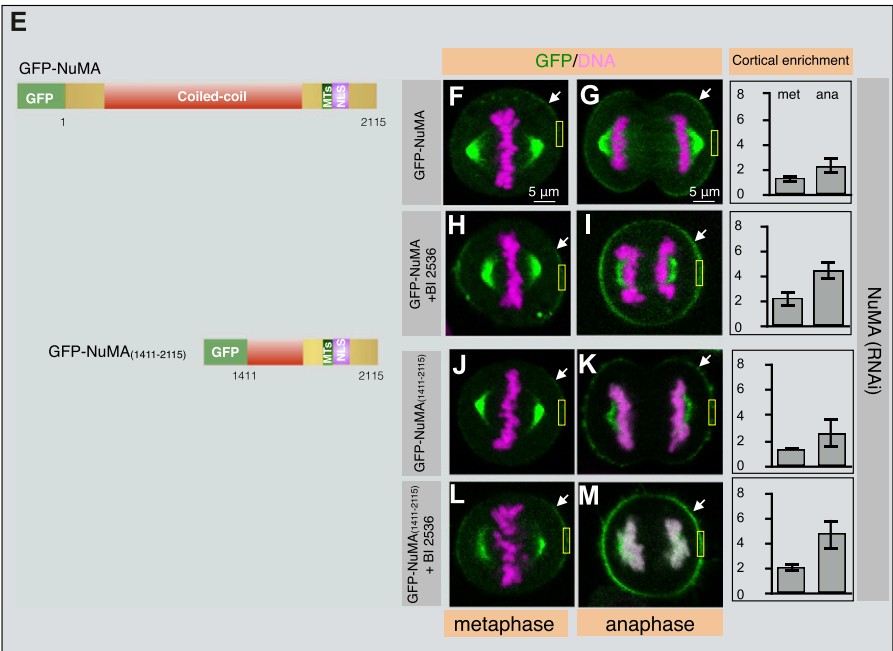

metaphase    anaphase

**Figure 1.    Cortical NuMA enrichment upon Plk1 inhibition is dynein independent.**

**(A–D)** HeLa cells in metaphase, as indicated, transfected with control siRNAs (A), and in addition treated with Plk1 inhibitor BI 2536 (300 nM; this regimen is used for most of the experiments, if not specified) for 30 min (B), transfected with siRNAs against DHC1 (C) and also treated with Plk1 inhibitor BI 2536 (D). Cells were fixed 72 h thereafter and stained for NuMA (green) and p150[Glued] (red). In this and other figures, DNA is visualized in blue and arrows point to cortical localization. 98% of control metaphase cells exhibited weak cortical NuMA and p150[Glued] staining, as shown, and NuMA cortical levels did not significantly change upon DHC1 (RNAi) treatment. Also, note the loss of p150[Glued] cortical signal in cells treated with siRNAs against DHC1. BI 2536–treated metaphase cells exhibited a strong cortical NuMA signal in 100% of control and DHC1 (RNAi) cells as shown (n > 150 cells for all cases were visually quantified). In this and other Figures, quantification of the cortical enrichment was performed in an area of size 1.8 × 4 $\mu$m (shown in yellow), see Materials and Methods section for details. Moreover, quantification of cortical enrichment is shown on the right for metaphase (met) for five cells in each condition; ($P$ < 0.0005 between control and BI 2536 for cortical NuMA levels and $P$ < 0.0005 between DHC1 [RNAi] and DHC1 [RNAi] cells that are also treated with BI 2536 for cortical NuMA levels; error bars: SD). **(E)** Schematic representation of NuMA constructs used for the experiments that is shown below; the coiled-coil domain, the regions mediating interaction with microtubules (MTs), and the nuclear localization signal (NLS) are represented. **(F–M)** Images from time-lapse microscopy of HeLa cells stably expressing mCherry-H2B and partly depleted of endogenous NuMA by RNAi using siRNAs sequences targeting 3'UTR of NuMA (see depletion efficiency of siRNAs in Fig S3D). These cells as indicated are transfected with GFP-NuMA (F, G) and also treated with BI 2536 (H, I) or transfected with GFP-NuMA$_{(1,411–2,115)}$ (J, K) and also treated with BI 2536 (L, M). The GFP signal is shown in green, the mCherry signal in pink. Acute treatment with BI 2536 in cells expressing GFP-NuMA and GFP-NuMA$_{(1,411–2,115)}$ cause substantial enrichment of cortical NuMA in metaphase and anaphase. Quantification of cortical enrichment is shown on the right for metaphase (met) and anaphase (ana) for 10 cells in each condition; see Materials and Methods section ($P$ < 0.0072 between metaphase and anaphase untreated cells, $P$ < 0.005 between metaphase or anaphase cells expressing GFP-NuMA either left untreated or treated with BI 2536, and $P$ < 0.005 between metaphase or anaphase cells expressing GFP-NuMA$_{(1,411–2,115)}$ which are either left untreated or treated with BI 2536; error bars: SD).

depletion of LGN resulted in the loss of cortical NuMA/p150[Glued] in cells which are either exposed to DMSO (control) or Plk1 inhibitor BI 2536 in metaphase (Fig 2A–D; see Fig S3A–C for depletion efficiency of LGN siRNAs). Similarly, we compared the impact of NuMA depletion on the cortical LGN levels in cells treated or untreated with BI 2536. Because NuMA and LGN are co-dependent for their cortical localization in metaphase (Du & Macara, 2004; Woodard et al, 2010; Kotak et al, 2014), cells treated with NuMA siRNAs showed a marked reduction in cortical LGN (compare Fig 2G with 2E; see Fig S3E for depletion efficiency of NuMA siRNAs). Moreover, BI 2536 treatment of NuMA (RNAi) cells also did not result in the cortical LGN localization (compare Fig 2H with 2F). Overall, our data indicate that NuMA and LGN cooperate with each other to promote their cortical enrichment upon Plk1 inhibition during metaphase.

To specify which proteins among NuMA or LGN could be a potential target of Plk1, we sought to assess NuMA localization in anaphase cells. In human cells, levels of NuMA/p150[Glued] dramatically increases in the polar cortical regions during anaphase, and

that this is in a manner independent of LGN (Kiyomitsu & Cheeseman, 2013; Seldin et al, 2013; Kotak et al, 2014; Zheng et al, 2014). Thus, we reasoned if cortical levels of NuMA enrich over wild-type during anaphase upon acute Plk1 inactivation, this must be independent of LGN. Indeed, we observed substantial enrichment in NuMA/p150[Glued] localization in BI 2536 treated anaphase cells (compare Fig 2J with 2I). Analogous results were obtained in cells expressing GFP-NuMA (compare Fig 1I with 1G) or GFP-NuMA$_{(1,411–2,115)}$ (compare Fig 1M with 1K). Acute inhibition of Plk1 in non-transformed hTERT1-RPE1 cells also caused excess cortical NuMA during anaphase (compare Fig S2E with S2C). Furthermore, partial depletion of Plk1 by RNAi also resulted in a significant increase in NuMA levels during anaphase (compare Fig S2M with S2L). Importantly, this cortical enrichment of NuMA during anaphase was also seen in anaphase cells depleted of LGN by RNAi (compare Fig 2L with 2K). To strengthen these findings further, we analyzed the metaphase and anaphase cortical localization of GFP-NuMA$_{(ΔLGN)}$ in cells depleted for endogenous NuMA in the presence and absence of BI 2536. GFP-NuMA$_{(ΔLGN)}$ lacks a motif

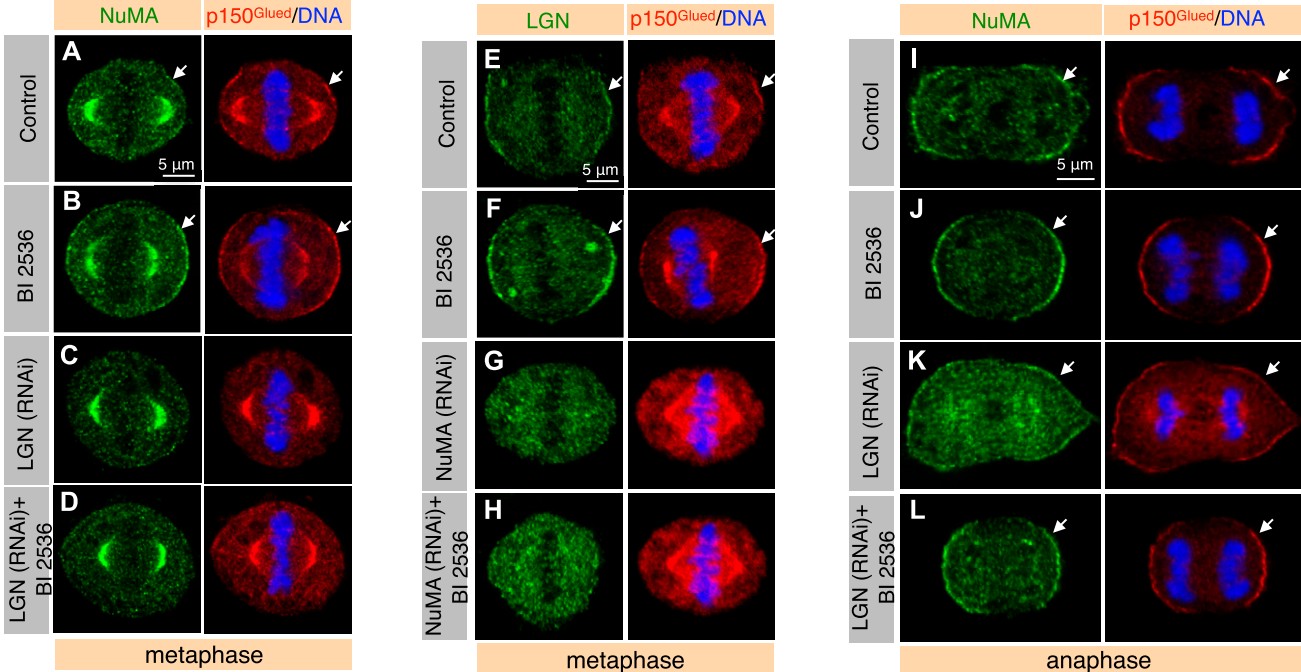

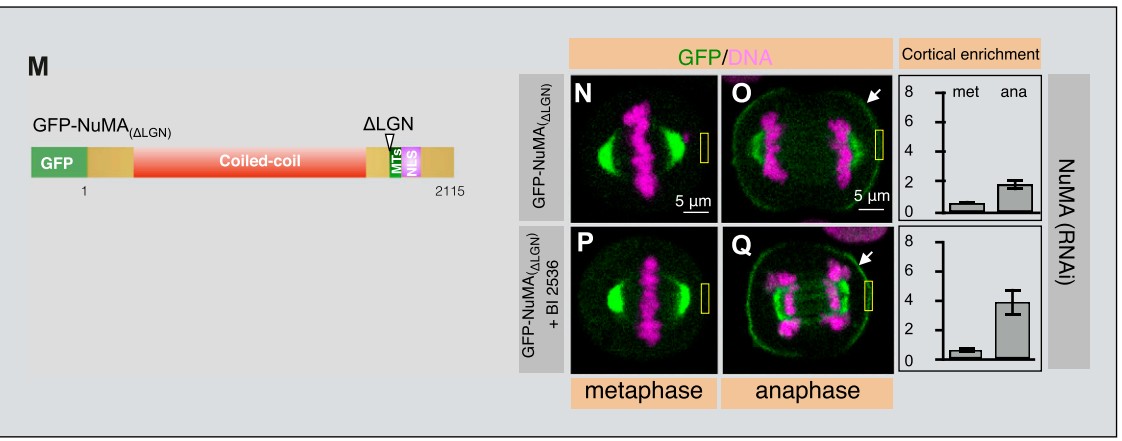

**Figure 2.   LGN is essential for Plk1-dependent cortical NuMA enrichment during metaphase, but not in anaphase.**
**(A–D)** HeLa cells in metaphase, as indicated, transfected with control siRNAs (A), and in addition treated with Plk1 inhibitor BI 2536 (B), transfected with LGN siRNAs (C) and also treated with Plk1 inhibitor BI 2536 (D). Cells were fixed 60 h post RNAi and stained for NuMA (green) and p150$^{Glued}$ (red). 98% of control metaphase cells exhibited weak cortical NuMA and p150Glued staining, and this is reduced to 1% in cells depleted for LGN. BI 2536-treated metaphase cells exhibited strong cortical NuMA and p150$^{Glued}$ signal in 100% of control cells and this is in contrast to the LGN (RNAi) cells treated with BI 2536 where only 2% of the cells exhibited cortical enrichment in NuMA and p150$^{Glued}$ signal (n > 200 cells for all cases and represented cell images are shown in the figure panels). See Fig S3 for RNAi-mediated depletion efficiency. **(E–H)** HeLa cells in metaphase, transfected with control siRNAs (E), and in addition treated with Plk1 inhibitor BI 2536 (F), transfected with NuMA siRNAs (G), and also treated with Plk1 inhibitor BI 2536 (H). Cells were fixed 60 h post RNAi and stained for LGN (green) as well as p150$^{Glued}$ (red). 100% of control metaphase cells exhibited weak cortical LGN and p150Glued staining, and this is reduced to 0% in cells depleted for NuMA by siRNAs. Metaphase cells that are depleted of NuMA by siRNAs and are exposed to BI 2536 exhibited no cortical LGN and p150Glued signal in 100% of the cells in contrast to the control cells that show robust LGN and p150$^{Glued}$ signal when treated with BI 2536. More than 100 cells were visually quantified, and the represented images are shown here. Also, see Fig S3 for RNAi-mediated depletion efficiency. **(I–L)** HeLa cells in anaphase as indicated, transfected with control siRNAs (I), and in addition treated with Plk1 inhibitor BI 2536 (J), transfected with LGN siRNAs (K), and also treated with Plk1 inhibitor BI 2536 (L). Cells were fixed 60 h post RNAi and stained for NuMA (green) as well as p150$^{Glued}$ (red). 100% of the control or LGN (RNAi) anaphase cells exhibited significant cortical NuMA and p150$^{Glued}$ staining in 100% of the cells and that cortical localization becomes stronger in cells acutely treated with BI 2536 in 100% of the cells (n > 50 cells for all cases). **(M)** Schematic representation of full-length GFP-NuMA constructs lacking LGN-binding motif at the C-terminus. The coiled-coil domain, the regions mediating interaction with microtubules (MTs), and the nuclear localization signal (NLS) are shown. **(N–Q)** Images from time-lapse microscopy of HeLa cells stably expressing mCherry-H2B and partly depleted of endogenous NuMA by RNAi using siRNAs sequences targeting 3'UTR of NuMA. As indicated, these cells are transfected with GFP-NuMA$_{\Delta LGN}$ (N, O) and also treated with BI 2536 (P, Q). The GFP signal is shown in green and the mCherry signal in pink. GFP-NuMA$_{(\Delta LGN)}$ does not localize at the cell cortex in metaphase in contrast to its anaphase cortical localization. Acute treatment with BI 2536 in cells expressing GFP-NuMA$_{\Delta LGN}$ cause robust enrichment of cortical NuMA in anaphase cells in contrast to the control anaphase cells. Quantification of cortical enrichment is shown on the right for metaphase (met) and anaphase (ana) for five cells in each condition; see Materials and Methods section ($P = 0.15$ for metaphase cells expressing GFP-NuMA$_{(\Delta LGN)}$ treated or untreated with BI 2536 during metaphase and $P < 0.0005$ anaphase cells for GFP-NuMA$_{(\Delta LGN)}$ treated or untreated for BI 2536; error bars: SD).

that is critical for interaction with LGN during metaphase (Du et al, 2001; Du & Macara, 2004; Kotak et al, 2014) (Fig 2M). We found that the cortical levels of GFP-NuMA$_{(\Delta LGN)}$ is undetectable during metaphase in cells either left untreated or treated cells with BI 2536 (Fig 2N and P). However, GFP signal remains associated with the cortex in anaphase (Fig 2O), and substantially augmented in anaphase cells upon addition of BI 2536 (Fig 2Q). These data strongly suggest that Plk1 negatively regulate cortical NuMA localization and that this impact of Plk1 on NuMA is presumably independent of LGN, at least in the anaphase cells.

### Alteration in NuMA turnover is responsible for its enrichment at the cell cortex in metaphase upon Plk1 inhibition

Plk1 inhibition enriches NuMA at the cell cortex in metaphase, and therefore, we first sought to determine if cortical NuMA enrichment stems from its reduced amount at the spindle pole or the cytosol or both. Interestingly, we uncovered that analogous to the cell cortex, spindle-pole levels of NuMA were markedly elevated and cyto-plasmic NuMA levels were relatively low in cells acutely treated with Plk1 inhibitor BI 2536 (Fig S4A–D), suggesting that overall decrease in the cytoplasmic levels of NuMA could be the reason for NuMA enrichment at the cell cortex and to the spindle poles. NuMA enrichment at the spindle poles was surprising to us, and thus, we wondered if this increase in levels of NuMA at the poles and the cell cortex is due to a change in the molecular dynamics of NuMA in BI 2536–treated cells. To address this, we initially conducted FRAP experiments at the spindle poles in cells expressing GFP-NuMA that are either left untreated or acutely treated with BI 2536. We found a dramatic increase in the half-time for the recovery ($t_{1/2}$) of GFP-NuMA at the poles in the BI 2536–treated cells in comparison with the untreated control cells (Fig S4E–H). This observation prompted us to conduct FRAP experiments at the cell cortex in cells ex-pressing GFP-NuMA. Notably, similar to the spindle poles, the re-covery time ($t_{1/2}$) for cortical GFP-NuMA as measured by FRAP was increased in Plk1-inhibited cells than in control cells ($t_{1/2}$ for GFP-NuMA = 6.7 s and $t_{1/2}$ for GFP-NuMA + BI 2536 = 16.0 s) (Figs 3A, B, D, and E). In addition, the mobile fraction of GFP-NuMA is markedly decreased at the cell cortex upon BI 2536 treatment. Because endogenous LGN is absent at the spindle poles (Kotak et al, 2012), and the half-time for the recovery ($t_{1/2}$) for GFP-NuMA are high both at the spindle poles and at the cell cortex in BI 2536–treated cells further indicating that Plk1-mediated modulation of NuMA, but not LGN, is the key for NuMA cortical enrichment upon BI 2536 treat-ment. In the same realm, we did not uncover any substantial change in the NuMA and LGN interaction in cells treated with BI 2536 in comparison with the control cells (Fig S4I). Overall, these data indicate that a modulation of NuMA turnover upon BI 2536 treat-ment is responsible for its cortical enrichment during metaphase.

### Plk1 inhibition impacts spindle orientation during metaphase

What is the biological significance of excess cortical NuMA during metaphase? It has been previously shown that high cortical levels of either the ternary complex component NuMA, LGN, or dynein can affect spindle orientation during metaphase (Du & Macara, 2004; Kiyomitsu & Cheeseman, 2012, 2013; Kotak et al, 2012; Kotak et al, 2014). Long-term incubation of Plk1 inhibitor perturbs bipolar

spindle formation, and thus, such cells are not refractory to analyze the impact of BI 2536 treatment on spindle orientation. Since our setup of Plk1 inhibition allows cells to retain bipolarity, we could analyze the effect of BI 2536 treatment on spindle orientation. To this end, we monitored spindle orientation by analyzing fixed cells grown on coverslips with an L-shaped fibronectin micro-patterns (Fig 4A; Théry et al, 2005) either acutely treated with DMSO (control) or BI 2536. We found that spindle orientation is perturbed in cells which are exposed to BI 2536 before fixation in contrast to the control condition (compare Fig 4C with 4B and 4E with 4D).

Next, we sought to investigate the impact of Plk1 inhibition on the spindle orientation in non-transformed hTERT1-RPE1 cells. hTERT1-RPE1 cells remain relatively flat in metaphase in contrast to the HeLa, and we discovered that in the DMSO-treated control cells, the distance between two spindle poles in the z-axis during metaphase remains close to or less than 1 $\mu$m in the majority of the cells (Fig 4F). However, cells that are acutely treated with BI 2536 show significant tilt in the mitotic spindle as measured by calculating the distance in z-axis between two spindle poles in metaphase (compare Fig 4G with 4F and 4I with 4H). Overall, these data suggest that the accumulation of excess cortical force generators (NuMA/ LGN/dynein) can impact spindle orientation in Plk1 inhibitor–treated metaphase cells.

### NuMA is phosphorylated by Plk1 at its C-terminus

As cortical NuMA enrichment upon Plk1 inhibition is independent of LGN in anaphase, we entertained the possibility of NuMA being the direct target of Plk1. To this end, we performed GST pull-down assays using recombinant PBDs (aa 305–603) of Plk1 made in *E. coli* (Fig 5A), and found that endogenous NuMA from the Nocodazole synchronized mitotic cells could efficiently interact with the GST-Plk1PBD, but not GST alone (Fig 5B). Similarly, GFP-NuMA$_{(1,411–2,115)}$ that robustly enriches at the cell cortex upon Plk1 inhibition (Fig 1L) interacts with GST-Plk1PBD, but not with GST (Fig 5C), suggesting that NuMA interacts with PBD of Plk1 through its C-terminus se-quences. Next, we plan to establish whether GFP-NuMA$_{(1,411–2,115)}$ also associates with endogenous Plk1 in a manner independent of LGN in a co-immunoprecipitation experiment. Interestingly, we found that GFP-NuMA$_{(1,411–2,115)}$ interacts with endogenous Plk1 in nocodazole-synchronized mitotic cell lysates lacking LGN (Fig 5D). Conversely, we also uncovered that GST-Plk1PBD interacts with endogenous NuMA in the mitotic cell extracts depleted of LGN (Fig S4J). These results together with the data from Figs 2 and 3 firmly establish that NuMA, but not LGN, is the target of Plk1.

PBD of Plk1 could interact with NuMA directly or indirectly; therefore, we investigate the possibility of a direct interaction between NuMA and Plk1PBD by performing far-Western blot anal-ysis. We immunoprecipitate GFP-NuMA$_{(1,411–2,115)}$ from the cell lysate of nocodazole-synchronized mitotic cells and conducted Far Western (FW) analysis either with GST or with GST-Plk1PBD. In-terestingly, we found that Plk1PBD interacts with immunoprecipi-tated GFP-NuMA$_{(1,411–2,115)}$ in an FW blot (Fig 5E). Altogether, these results strongly suggest that Plk1 can directly associate with NuMA in mitotic cells.

NuMA is highly phosphorylated during mitosis (Price & Pettijohn, 1986; Compton & Luo, 1995; Sparks et al, 1995; Saredi et al, 1997) and

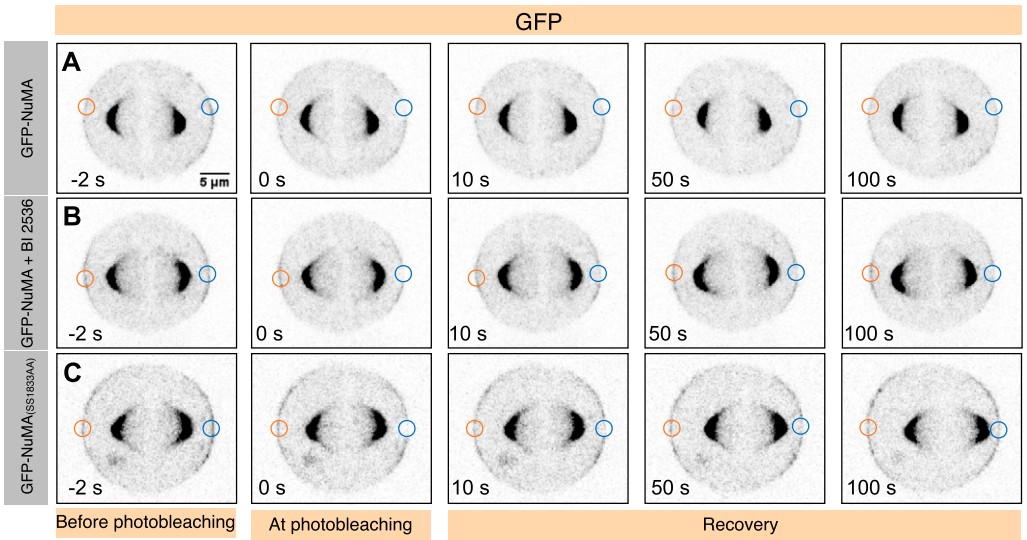

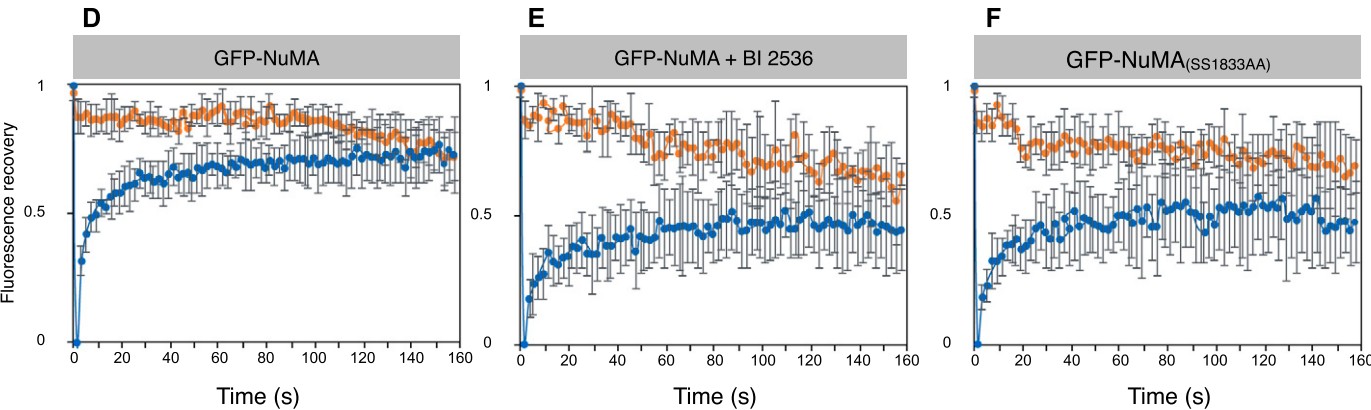

**Figure 3. Plk1 inhibition affects the turnover of NuMA at the cell cortex.**
**(A–F)** FRAP analysis of HeLa cells stably expressing mCherry-H2B at metaphase and transfected with GFP-NuMA (A, D), also treated with BI 2536 (B, E) or transfected with GFP-NuMA(SS1833/34AA) (see Materials and Methods section for details). In the inverted confocal images, the GFP signal is shown in grey. The unbleached and bleached region of the cell cortex is shown by orange and blue circles, respectively. The recovery profile of the bleached cortical signal is plotted for 160 s for all three conditions. Note slow ($t_{1/2}$) of GFP-NuMA cells which are either treated with BI 2536 (16.0 s) or expressing GFP-NuMA(SS1833/34AA) (12.4 s) in comparison with the control condition (6.7 s). Also note that the mobile fraction of GFP-NuMA is also significantly affected in cells which are either acutely treated with BI 2536 or expressing GFP-NuMA(SS1833/34AA). $P < 0.05$ between control (DMSO) and BI 2536–treated cells, and control (DMSO) and GFP-NuMA(SS1833/34AA)–expressing cells as calculated by two-tailed $t$ test (n > 10 cells for all cases; error bars: SD).

its phosphorylation is regulated by various kinases including Cdk1 and Aurora A (Compton & Luo, 1995; Kiyomitsu & Cheeseman, 2013; Kotak et al, 2013, 2016; Seldin et al, 2013; Zheng et al, 2014; Gallini et al, 2016). It was shown earlier that Cdk1 phosphorylation creates the docking sites on the metaphase binding partners for Plk1 (Elia et al, 2003a, b). However, since Cdk1 is inactive during anaphase and acute inactivation of Plk1 with BI2536 during anaphase causees robust cortical enrichment of NuMA, this appears to be an unlikely mechanism for the creation of a Plk1 docking site on NuMA. Therefore, we tested the possibility of NuMA being directly phosphorylated by Plk1. Importantly, we found that recombinant hexa histidine-tagged C-terminus of NuMA after its coiled-coil region [6His-NuMA(1,700–2,115); Fig 5A] can be phosphorylated by Plk1 in an in vitro kinase assay (Fig 5F). Overall, these data clearly indicate that that Plk1 acts as a bona fide kinase that interacts and phosphorylates NuMA which may impact its cortical localization both during metaphase and anaphase.

### Phosphorylation of NuMA by Plk1 at 1833/34 residue can impact its cortical localization

Because the C-terminus of NuMA after the big coiled-coil [NuMA(1,700–2,115)] gets in vitro phosphorylated by Plk1, we checked the potential Plk1-phosphorylation sites on this fragment. This analysis was partly based on the presence of putative Plk1 consensus such as [D/E]Xp[S/T][FLIYWVM] (Nakajima et al, 2003), [NXp(S/T)] or [p(S/T)F] (Kettenbach et al, 2011) on NuMA(1,700–2,115) and also on the loss of phosphorylation on these sites upon BI 2536 treatment in a genome-wide quantitative phosphoproteomics data (Kettenbach et al, 2011). These approaches led to the identification of four

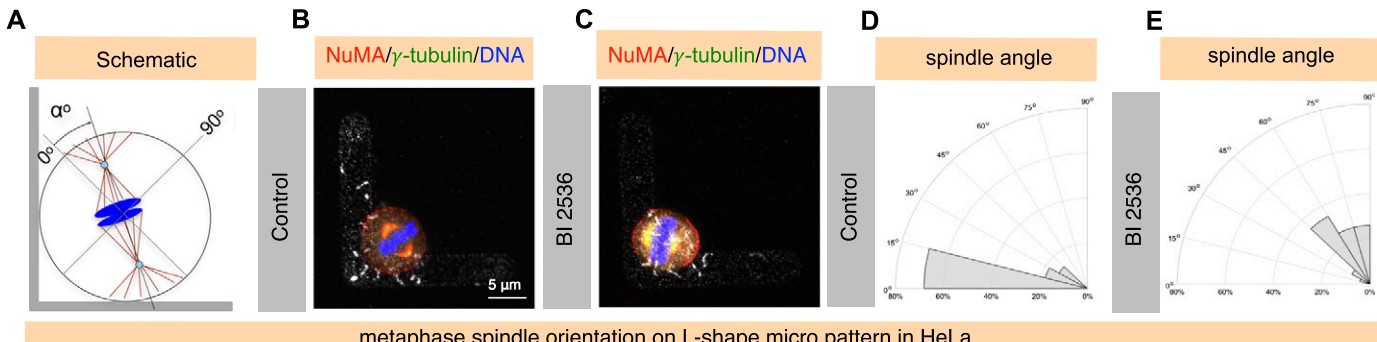

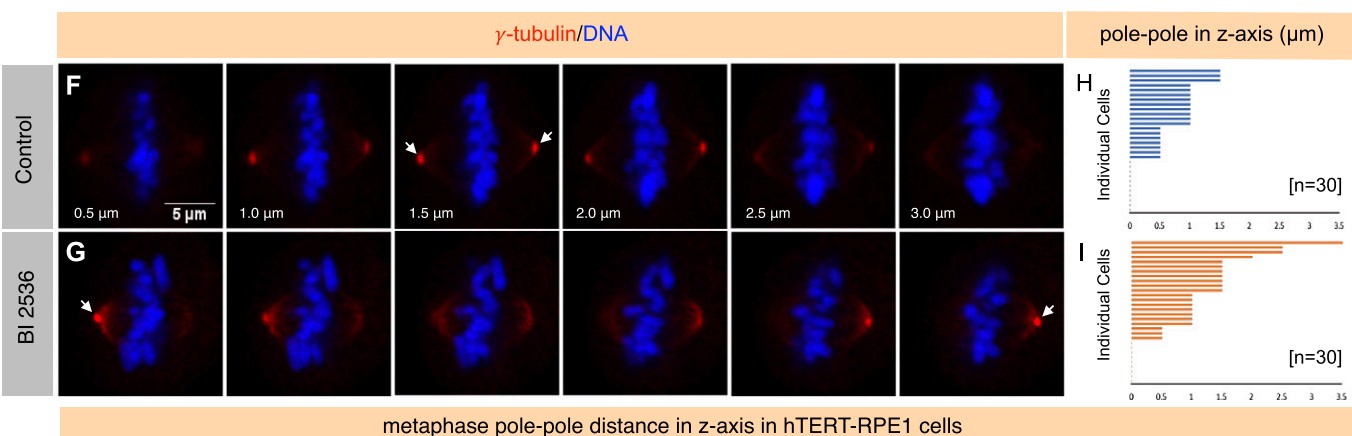

**Figure 4. Acute Plk1 inhibition impacts mitotic spindle orientation.**
**(A)** Schematic representation of the mitotic spindle on an L-shaped micro-pattern. The spindle is shown in brown, and centrosomes in cyan and chromosomes in blue. Spindle orientation was determined as an angle as depicted, with 0° being defined as the position of spindle along the hypotenuse as shown. **(B–E)** Synchronized HeLa cells on a L-shaped fibronectin micropattern either treated with DMSO (B, D) or BI 2536 (C, E). Cells are stained for NuMA (red) and γ-tubulin (green). DNA is shown in blue. D and E represent the frequency of angular distribution of spindle positioning every 15°. Note the change in the axis of the spindle orientation in cell treated with Plk1 inhibitor BI 2536 in contrast to the control condition. Also note cortical NuMA enrichment in cell treated with BI 2536 (N = 25 for DMSO control and N = 27 for BI 2536). $P < 0.005$ between DMSO and BI 2536 treated cells, two-tailed $t$ test. **(F–I)** hTERT-RPE1 cells were treated with DMSO control (F) or BI 2536 for 30 min post 7.5 h release of double thymidine (G). Cell were stained for γ-tubulin (red). DNA is shown in blue. The arrow represents the brightest γ-tubulin signal in the z-axis. Please note the metaphase spindle is tilted for a cell treated with Plk1 inhibitor BI 2536 in contrast to control. **(H, I)** represents pole–pole distance in the z-axis for control (H) or for BI 2536 (I) treated metaphase cells. Individual cells are plotted on the y-axis, and the distance between spindle pole is shown in $\mu m$ on the x-axis. Please note that a few cells in the control and those treated with BI 2536 are on the same z-plane and thus show 0 $\mu m$ distance on the z-axis (shown by dots) between spindle-poles (N = 30 for both conditions). $P < 0.05$ between DMSO control and BI 2536–treated cells, two-tailed $t$ test.

potential Plk1 phosphorylation sites on NuMA, namely S1789, T1818, S1830, and SS1833/34 (both of these Serine residues get phosphorylated in the phosphoproteomics dataset; Kettenbach et al, 2011). Next, we set out to investigate the physiological relevance of these residues for cortical NuMA localization. Because GFP-NuMA$_{C-ter}$ (aa 1,411–2,115) behaves as a full-length protein with respect to its cortical localization (Kotak et al, 2012, 2013) (Fig 1H and I), we expressed either wild-type GFP-NuMA$_{C-ter}$ or mutated fragments where the abovementioned Threonine (T) or Serine (S) residues were mutated to Alanine (A) and examined the cortical GFP signal in metaphase cells by performing a time-lapse live recording (Fig 6A–E). Importantly, GFP-NuMA$_{C-ter(SS1833/34AA)}$ showed significantly enriched cortical levels in comparison with three other mutated fragments (compare Fig 6B with 6A and 6C–6E). In addition, we found that the full-length GFP-NuMA$_{(SS1833/34AA)}$ also enriches significantly more to the cell cortex during metaphase and anaphase in comparison with the wild-type GFP-NuMA with or without endogenous NuMA (compare Fig 6G with 6F and data not shown).

Because GFP-NuMA(SS1833/34AA) significantly enriches at the cell cortex, we decided to examine if this could be due to change in mobility of GFP-NuMA$_{(SS1833/34AA)}$ at the cell cortex. Notably, we found that the recovery time ($t_{1/2}$) for cortical GFP-NuMA(SS1833/34AA) during metaphase as measured by FRAP was significantly longer in comparison with GFP-NuMA ($t_{1/2}$) for GFP-NuMA = 6.7 s and $t_{1/2}$ for GFP-NuMA(SS1833/34AA) = 12.4 s; compare Fig 3C and F with 3A and D).

Enriched levels of cortical NuMA are implicated in spindle misorientation (Kotak et al, 2013), and therefore, we attempted to analyze the impact of SS1833/34AA mutations on spindle orientation. To this end, we conducted spindle orientation experiments by using L-shaped micro-patterns (Fig 6H). Intriguingly, in contrast to the control cells or cells expressing GFP-NuMA, mild expression of GFP-NuMA$_{(SS1833/34AA)}$ causes a significant impact on spindle orientation on L-shaped micro-patterns (compare Fig 6K with 6J or 6I). Altogether, these data support the notion that SS1833/34AA serves as a putative Plk1 phosphorylation site on NuMA and the phosphorylation of this

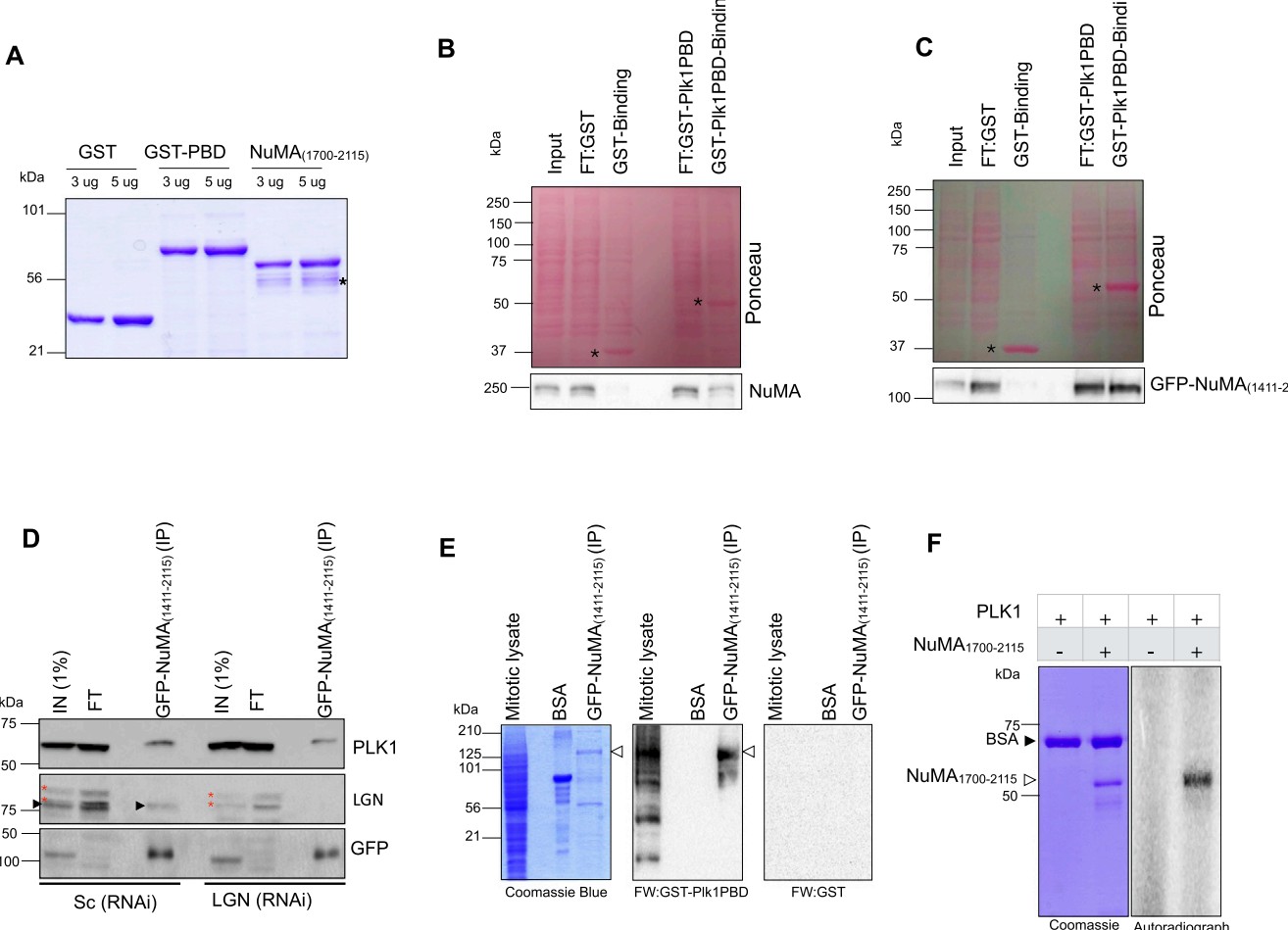

**Figure 5. NuMA directly interacts with Plk1 and gets phosphorylated at its C-terminus.**
**(A)** Recombinantly purified GST, GST tagged Polo-Box Domain (PBD) of Plk1 (aa 305–603), or Hexa-histidine tagged NuMA(1,700–2,115) run on SDS–PAGE and stained with Coomassie Blue. In this and other panels, the molecular mass is indicated in kilodaltons (kD). Note that bacterially expressed NuMA(1,700–2,115) is unstable, thus explaining the presence of few other species (marked by an asterisk). **(B)** GST pull-down assay using recombinantly purified GST or GST-tagged Plk1-PBD in Nocodazole-synchronized mitotic cells extracts. Asterisk on the Ponceau-stained nitrocellulose membrane depicts GST alone or GST-tagged PBD of Plk1. Note Plk1-PBD pull-down endogenous NuMA as detected by Western blotting using NuMA-specific antibodies, in contrast to the GST alone. FT:GST represents FT of the GST fraction and FT: GST-Plk1PBD represents FT of the GST-Plk1PBD fraction. Input is 1% of the total pull-down fraction (3 mg) and the binding fraction is one-third of the total binding reaction (~1 mg). **(C)** Similar to the above, the GST pull-down assay using recombinantly purified GST or GST-tagged Plk1-PBD with the Nocodazole synchronized mitotic cells expressing GFP-NuMA(1,411–2,115). Asterisk on the Ponceau stained nitrocellulose membrane depicts GST or GST-tagged PBD of Plk1. Note Plk1-PBD pull-down GFP-NuMA(1,411–2,115) as detected by Western blotting using GFP-specific antibody, in contrast to the GST alone. FT: GST represents the FT of the GST fraction and FT: GST-Plk1PBD represents the FT of the GST-Plk1PBD fraction. Input is 1% of the total pull-down fraction (3 mg) and the binding fraction is one-third of the total binding reaction (~1 mg). **(D)** Immunoprecipitation (IP) using GFP-trap from Nocodazole-arrested mitotic HeLa cells extracts which are either non-depleted or depleted of LGN by RNAi as indicated and also expressing GFP-NuMA(1,411–2,115). Resulting blots were probed for Plk1 and LGN antibodies as indicated. IN (1% of total), IP: 50% of the total. Please note that for the GFP detection in the IP fraction, only 5% of the total IP fraction was loaded. Note that Plk1 interacts with GFP-NuMA(1,411–2,115) irrespective of the presence and absence of LGN. Asterisks represent non-specific bands detected with LGN antibodies. **(E)** Immunoprecipitation (IP) with GFP-Trap from nocodazole-arrested mitotic HeLa cells extracts expressing GFP-NuMA(1,411–2,115), total mitotic lysate, and BSA were probed for Far-Western (FW) using either GST-Plk1PBD or with GST alone. Western blotting was performed for GST antibodies. Please note that GST-Plk1PBD, but not GST, interacts with the IP fraction of GFP-NuMA(1,411–2,115) and mitotic lysate but not with the purified BSA. Also, note that Plk1PBD interacts with several mitotic substrates in the mitotic lysates. **(F)** In vitro kinase assay with the recombinantly purified Hexa-histidine tagged NuMA(C-ter) (aa 1,700–2,115) incubated with recombinant Plk1 generated in sf21 insect cells plus [γ-32P]-ATP and analyzed by autoradiography (right), on the left, equal loading is shown by Coomassie staining, BSA serves as a negative control. FT, flow through.

residue impacts cortical NuMA levels possibly by modulating NuMA turnover and thus impacting proper spindle orientation.

Previous reports have implicated Plk1 in regulating spindle behavior by modulating dynein levels (Collins et al, 2012; Kiyomitsu & Cheeseman, 2012). Recently, it was suggested that the impact of spindle-pole Plk1 on cortical dynein occurs either at the LGN level or at a level that is upstream of LGN (Tame et al, 2016); however,

what is the direct-target of Plk1 remained elusive. NuMA and LGN are co-dependent on each other for their cortical localization in metaphase (Du & Macara, 2004; Woodard et al, 2010; Kiyomitsu & Cheeseman, 2012; Kotak et al, 2012), but not during anaphase (Kiyomitsu & Cheeseman, 2013; Seldin et al, 2013; Kotak et al, 2014; Zheng et al, 2014). Because Plk1 inactivation robustly enriches cortical NuMA, both in metaphase as well during anaphase, our

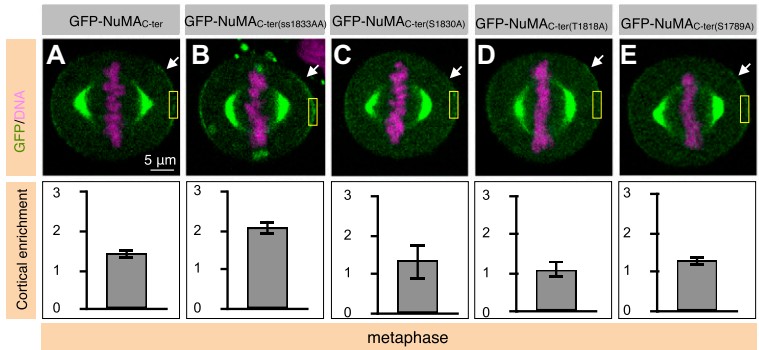

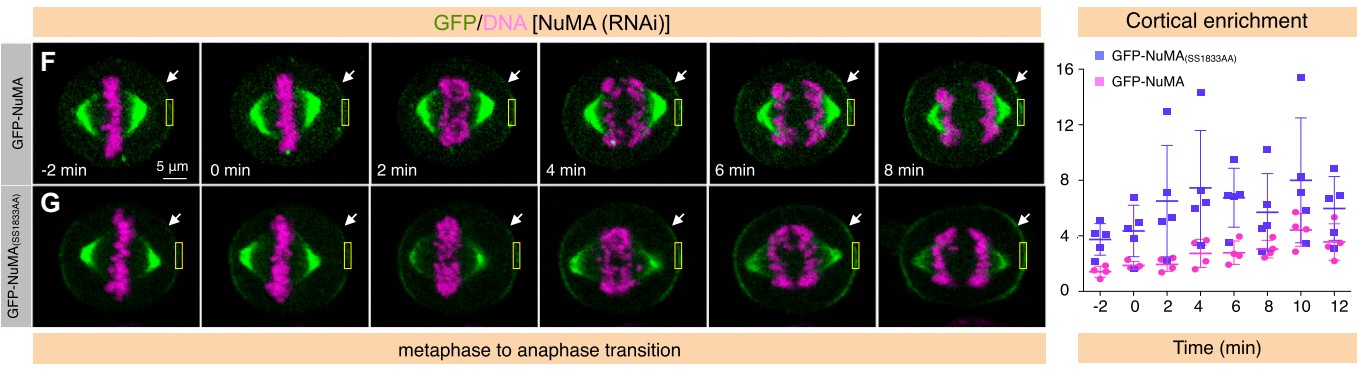

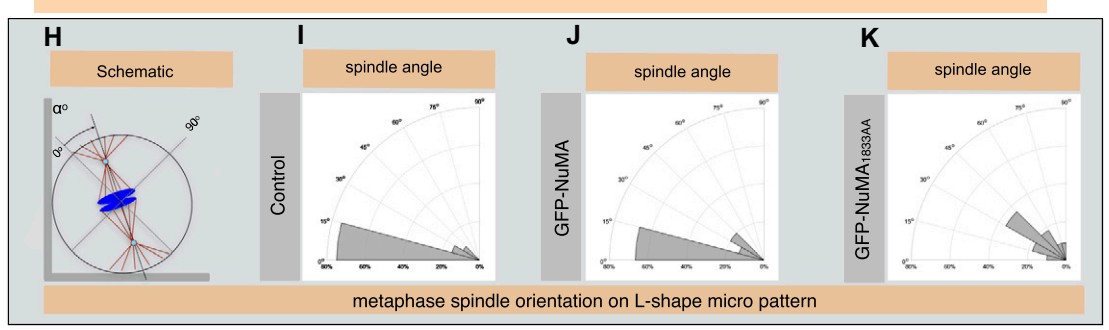

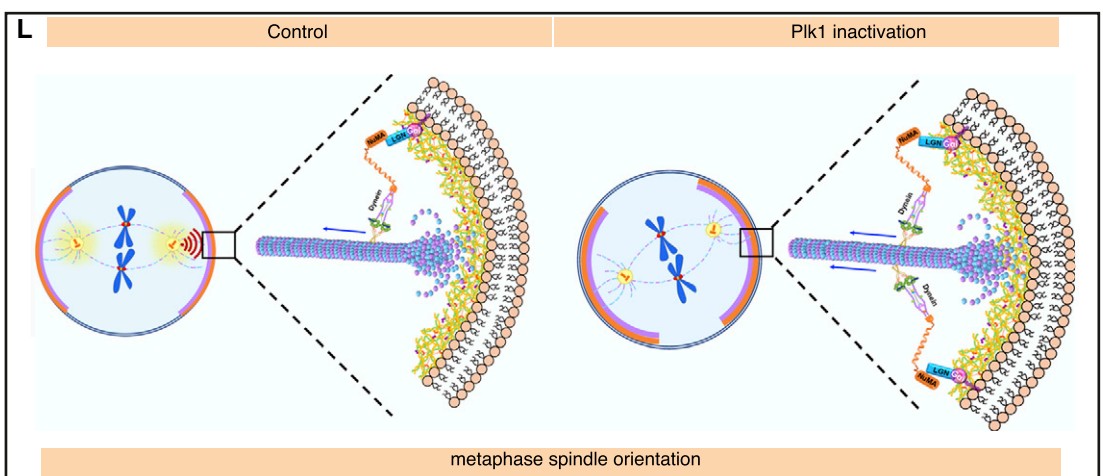

**Figure 6. Plk1 negatively regulates NuMA cortical localization by phosphorylating at its C-terminus.**
**(A–E)** Images from time-lapse microscopy of HeLa cells stably expressing mCherry-H2B at metaphase, as indicated, transfected with various mutant forms of C-terminus (1,411–2,115) of NuMA (see text for details). The GFP signal is shown in green and the mCherry signal in pink. Please note significant enrichment in the GFP signal in cells

work strongly supports the notion that Plk1 regulation occurs at the levels of NuMA (Fig 6L). We further reveal that the significantly reduced mobility of GFP-NuMA upon Plk1 inhibition or of GFP-NuMA$_{(SS1833/34AA)}$ at the cell cortex is presumably responsible by cortical NuMA enrichment in metaphase and possibly in anaphase cells. It would be interesting to uncover in the future whether this change in NuMA turnover by Plk1 is due to a direct impact of Plk1-mediated phosphorylation on NuMA structure or due to an alteration of its interaction with its binding partners such as F-actin–binding 4.1 family of proteins that is important for its stabilization at the cell cortex (Seldin et al, 2013). Intriguingly, phosphomimetic mutants for 1833/34 [GFP-NuMA(SS1833/34DD) or GFP-NuMA(SS1833/34EE)] behave as phospho-dead in their cortical localization (data not shown), and this could simply be because the phosphomimetic substitutions often fail to give the same outcome as achieved by protein phosphorylation (reviewed in Dephoure et al [2013]).

In addition to Plk1, Cdk1/cyclinB and Aurora A kinase were also shown to impact cortical NuMA and therefore the dynein localization during metaphase (reviewed in Seldin & Macara [2017]). This led to the question, why cortical NuMA acts as a potential target for most of these mitotic kinases? Because, NuMA mediates cortical localization of dynein during metaphase and in anaphase, one interesting possibility is that NuMA could function as a strategic molecule to orchestrate the accurate cortical localization of dynein for unperturbed mitosis in human cells.

Notably, Plk1 plays a myriad of functions in controlling various aspects of mitosis, and moreover, Plk1 overexpression is linked with tumorigenesis (Eckerdt et al, 2005; reviewed in Strebhardt & Ullrich [2006]). Because of the existence of some evidence that links spindle misorientation with tumorigenesis (Caussinus & Gonzalez, 2005; Quyn et al, 2010; Hehnly et al, 2015; Noatynska et al, 2012), it would be interesting for future work to evaluate if Plk1 driven cancer progression is due to its impact on spindle misorientation.

# Materials and Methods

### Cell culture, cell synchronization, spindle orientation, and transfection

HeLa cells expressing GFP-Centrin 1 (Piel et al, 2000), HeLa Kyoto, and hTERT1-RPE1 cells were cultured in high-glucose DMEM with GlutaMAX (CC3004; Genetix) media supplemented with 10% FCS in a humidified 5% $CO_2$ incubator at 37°C. For monitoring spindle positioning in fixed specimens, HeLa cells were synchronized with 2 mM thymidine (T1895; Sigma-Aldrich) for 20 h, released for 9 h, trypsinized, and plated on fibronectin L-shaped micropatterns (CYTOO SA). ~80,000 cells were placed on a CYTOO chip in a 35-mm culture dish. After 1 h, cells that had not attached to the micro-patterns were removed by gently washing with media. Cells were then fixed, 9 h after the release, with cold methanol and stained with antibodies against γ-tubulin (GTU88; Sigma-Aldrich) and NuMA (sc-48773; Santa Cruz). For Plk1 inhibition, cells were treated with 300 nM BI 2536 (S1109; Selleckchem) for 30 min on fibronectin L-shape micropatterns before fixation with cold methanol. To study spindle orientation in the z-axis using hTERT1-RPE1 cells, cells were synchronized using double thymidine, released for 7.30 h, and treated with BI 2536 for 30 min before fixation using cold methanol and staining with antibodies against γ-tubulin (GTU88; Sigma-Aldrich). Pole–pole distance in the z-axis was calculated by taking confocal z-sections of 0.5 μm covering both the brightest γ-tubulin signal intensity at the spindle-pole. The distance in z-axis was plotted for all the individual cells.

For siRNA experiments, ~100,000 cells were seeded in six-well plates. 6 μl of 20 μM siRNAs in 100 μl RNase-free water and 4 μl of Lipofectamine RNAiMAX (13778150; Invitrogen) in 100 μl RNase-free water were incubated in parallel for 5 min, mixed for 20 min, and then added to 2.5 ml medium per well. For plasmid transfections, cells were seeded at 80% confluency. 4 μg of plasmid DNA in 400 μl serum-free media either with 6 μl of either Turbofect (R0531; Thermo Fisher Scientific) or 4 μl Lipofectamine 2000 (11668019; Life Technologies) incubated for 15–20 min and added to each well.

### Plasmids and RNAi

All NuMA clones were constructed using full-length NuMA as a template with appropriate PCR primer pairs (the sequences of all primers are available upon request). The amplified products were sub-cloned into pcDNA3-GFP (Merdes et al, 2000). For recombinant expression in *E. coli*, PBD of Plk1 was cloned in pETEM 30 (EMBL-vectors) containing GST-tag at the N-terminus with the restriction sites NcoI and EcoRI. NuMA1700-2115 was cloned in pET28a plasmid with HexaHistidine tag at the N-terminus with the restriction sites NotI and EcoR1.

expressing GFP-NuMA$_{C-ter(SS1833/34AA)}$ (B) in contrast to cells expressing either a wild-type NuMA$_{C-ter}$ construct or the other mutant forms. Quantification of cortical enrichment is shown below for 10 cells in each condition; ($P < 0.005$ for metaphase cells expressing GFP-NuMA$_{C-ter(SS1833/34AA)}$ in comparison with either GFP-NuMA$_{C-ter}$ or other mutant constructs as analyzed by two-tailed *t* test; error bars: SD). **(F, G)** Images from time-lapse microscopy of HeLa cells stably expressing mCherry-H2B and partly depleted of endogenous NuMA at metaphase or at various stages of anaphase, as indicated, transfected either with GFP-NuMA (F) or GFP-NuMA$_{(SS1833/34AA)}$ (G). The GFP signal is shown in green and the mCherry signal in pink. Please note significant enrichment in the GFP signal in a cell expressing GFP-NuMA$_{(SS1833/34AA)}$ in contrast to the wild-type GFP-NuMA. Quantification of the cortical enrichment shown on the right representing the cortical signal as cells are progressing from metaphase to anaphase transition. For cells undergoing metaphase to anaphase transition from –2 min to 6 min, the $P < 0.05$ for cortical GFP signal between GFP-NuMA and GFP-NuMA$_{(SS1833/34AA)}$ and $P ≥ 0.1$ for 8 to 12 min intervals between GFP-NuMA and GFP-NuMA$_{(SS1833/34AA)}$, as analyzed by two-tailed *t* test; error bars: SD). **(H–K)** Schematic representation of the mitotic spindle on an L-shaped micro-pattern as shown in Fig 3A. Spindle orientation was determined as an angle as depicted, with 0° being defined as the position of spindle along the hypotenuse as shown. **(H–K)** represents the frequency of angular distribution of spindle positioning every 15°. Note the change in the axis of the spindle orientation in cells weakly expressing GFP-NuMA$_{(SS1833/34AA)}$ (K) in contrast to the untransfected control cells (I) or cells weakly expressing GFP-NuMA (J) [N = 20 for untransfected, N = 15 for cells expressing GFP-NuMA, and N = 23 for cells expressing GFP-NuMA$_{(SS1833/34AA)}$]. $P < 0.005$ between GFP-NuMA$_{(SS1833/34AA)}$–expressing cells and untransfected control cells, or between GFP-NuMA and GFP-NuMA$_{(SS1833/34AA)}$–expressing cells as calculated using two-tailed *t* test. **(L)** Model illustrating the impact of Plk1 on the cortical localization of NuMA and therefore dynein during metaphase. In control metaphase cells (on the left), spindle-pole pool of Plk1 (shown in yellow on the spindle-pole) creates a gradient (in red) and influences cortical dynein complex localization through phosphorylating NuMA (shown in orange). This process orchestrates the balanced levels of cortical levels of NuMA/dynein, which is essential for proper spindle orientation. Plk1 inactivation in metaphase (shown on the right), causes loss of negative regulation on NuMA and thus NuMA/dynein levels augment which causes to spindle orientation defects.

Double stranded siRNAs oligonucleotides were synthesized with the sequences: 5′-UAGGAAAUCAUGAUCAAGCAA-3′ (LGN-siRNAs; Life Technologies), [5′-GAACUAACAGCACGACUUA-3′, 5′-CUUCAGGGAUG-CAGUUAUA-3′, 5′-ACAGUGAAAUUCUUGCUAA-3′ and 5′-UGAAGGGU-UCUUUGACUUA-3′] (LGN-siRNAs, Dharmacon Smartpool On target plus), [5′-GGUGGCAACUGAUGCUUUA-3′, 5′-GAACCAGCCUCACCUAUC U-3′, 5′-GCAAACGGGUCUCCCUAGA-3′ and 5′-GGAGUUCGCUACCCU-GCUA-3′] (NuMA-siRNAs, Dharmacon Smartpool On target plus), [5′-GAUCAAACAUGACGGAAUU-3′, 5′-CAGAACAUCUCACCGGAUA-3′, 5′-GAAAUCAACUUGCCAGAUA-3′, 5′-GCAAGAAUGUCGCUAAAUU-3′] (DHC1-siRNAs, Dharmacon Smartpool On target plus), [5′-UA-GAACCCACACCCGAACAUGUACA-3′] (Plk1 siRNAs, Eurogentec).

In the cases where wild-type or mutant NuMA fusion constructs were expressed in NuMA-depleted cells, endogenous NuMA was depleted using siRNAs sequences CCUCUGGAUCUAGAAGGGACC-AUAA targeting 3′UTR sequence of NuMA that is missing in the fusion constructs.

### Plk1 inhibition using BI 2536

HeLa cells were treated with 2 mM thymidine for 23 h followed by 6-h release in media. Then, the cells were treated with 50 ng/ml Nocodazole (M1404; Sigma-Aldrich) for 5 h following which the cells were treated with 10 mM MG132 (M8699; Sigma-Aldrich) for 2 h. Cells were then released in media for 20 min. After this point, the cells were either treated with DMSO or 100 nM BI 2536. In the case of unsynchronized cells, 300 nM of BI 2536 was used to inhibit Plk1 for 30 min before fixation.

### Indirect immunofluorescence and time-lapse imaging of HeLa cells

For immunofluorescence, cells were fixed in −20°C methanol for 7–10 min and washed in PBS-0.05% Triton X-100 (PBST). After blocking in 1% BSA (RM3159; HiMedia) in PBST for 1 h, cells were incubated with primary antibodies for 4 h. Following three washes in PBST for 5 min each, cells were incubated with secondary antibodies for 1 h, counterstained with 1 $\mu$g/ml Hoechst 33342 (B2261; Sigma-Aldrich), washed three times for 5 min in PBST and mounted using Fluoromount-G (SouthernBiotech, 0100-01). Primary antibodies were used at the following dilutions: 1:200 rabbit anti-NuMA (sc-48773; Santa Cruz), 1:200 rabbit anti-LGN (HPA007327; Sigma-Aldrich), 1:200 mouse anti-p150Glued (612709; Transduction Laboratories), 1:2000 mouse anti-γ-tubulin (GTU88; Sigma-Aldrich), and 1:500 mouse anti-GFP (DsHB, 8H11s). Secondary antibodies used were 1:500 Alexa flour 488 goat anti-mouse (A11001; Invitrogen), 1:500 Alexa flour 488 goat anti-rabbit (A11008; Invitrogen), 1:500 Alexa flour 568 goat anti-mouse (A11004; Invitrogen), and 1:500 Alexaflour 568 goat anti-rabbit (A11011; Invitrogen). Confocal images were acquired on an Olympus FV 3000 confocal laser scanning microscope using a 60× NA 1.4 oil objective and processed in ImageJ and Adobe Photoshop, maintaining relative image intensities.

Time-lapse microscopy was conducted on an Olympus FV 3000 confocal laser scanning microscope using a 40× NA 1.3 oil (Olympus) using an imaging dish (0030740017; Eppendorf) at 5% $CO_2$, 37°C, 90%

humidity. Images were acquired every 3 min or 2 min, capturing 8–10 sections, 3 $\mu$m apart at each time point. Time-lapse figures were obtained using a single confocal section of the Z-stack.

### Quantification of cortical intensity

Quantification of cortical NuMA signal or GFP cortical signal in the case of GFP-tagged proteins for all the Figures was determined by calculating the ratio of the mean intensity of cortical signal (of an area of size 1.8 × 4 $\mu$m as shown in each Figure) divided by the mean intensity value in the cytoplasm (similar area) and correcting for the background signal (an analogous area outside the cell). The brightest polar cortical region was utilized as a selection criterion in control and in a given experimental condition. Significance was determined using two-tailed $t$ test for each condition.

### Quantification of spindle pole intensity

Quantification of spindle-pole GFP-NuMA signal was determined by calculating the ratio of the mean intensity of cortical signal (of an area of size 1.6 $\mu$m$^2$ as shown in each Figure) divided by the mean intensity value in the cytoplasm (similar area) and correcting for the background signal. The brightest spindle pole in each cell is utilized for the quantification. Significance was determined using two-tailed $t$ test for each condition.

### Quantification of cytoplasmic intensity

Quantification of cytoplasmic GFP-NuMA signal was determined by calculating the mean intensity of the cytoplasmic signal (of an area of size 1.8 × 4 $\mu$m as shown in each Figure) and correcting for the background signal. Significance was determined using two-tailed $t$ test for each condition.

### FRAP analysis

HeLa cells stably expressing mCherry-H2B were transfected with GFP-NuMA or GFP-NuMA(SS1833/34AA). FRAP experiments were performed for a specific region (4.85 $\mu$m$^2$ for the cell cortex and 17 $\mu$m$^2$ for the spindle pole with the 40× objective; as shown in Figs 3 and S4). 40% of the 488-nm laser was used to bleach the region of interest and images within the same focal plane were acquired for every 2 s for the cell cortex for the entire duration of 80 cycles or 5 s for the spindle pole for 50 cycles to monitor fluorescence recovery. For spindle pole FRAP, images were acquired over three planes (step size = 2 $\mu$m, thickness = 6 $\mu$m) because the spindle poles were going in and out of the imaging plane and the recovery was assessed from the maximum intensity projected images. For Plk1 inhibition, cells were treated with 300 nM BI 2536 for 30 min before FRAP experiments. To assess the fluorescence loss due to imaging-induced photobleaching, fluorescence from a cortical region or spindle pole separated from the bleached region was simultaneously recorded. The intensity value in the bleached area was measured, corrected for the background, and the curves were then normalized using the following equation:

$$I = \frac{(I_t - I_{min})}{(I_{max} - I_{min})}$$

where, I is the normalized intensity, $I_t$ is the intensity at a time-point, $I_{min}$ is the minimum intensity (at the time of bleaching), and $I_{max}$ is the maximum intensity (prebleaching intensity). For the calculation of half-time of recovery ($t_{1/2}$), the bleaching due to imaging was considered, and the values were quantified by fitting to an exponential equation.

### Immunoprecipitation, GST pull-down assays, and immunoblotting

For co-immunoprecipitations, 3 mg of cell lysate from the Nocodazole (100 nM) arrested mitotic HeLa cells was incubated with 30 $\mu$l GFP-Trap agarose beads (ACT-CM-GFA0050; ChromoTek) in lysis buffer (50 mM Tris, pH-7.4, 2 mM EDTA, 2 mM EGTA, 25 mM Sodium fluoride, 0.1 mM sodium orthovandate (S6508; Sigma-Aldrich), 0.1 mM PMSF (7110; Calbiochem), 0.2% Triton-X100, 0.3% NP-40, 100 nM Okadaic acid, and complete EDTA-free protease inhibitor (539134; Merck) for 4 h at 4°C. After extensive washing in wash buffer (50 mM Tris, pH-7.4, 2 mM EDTA, 2 mM EGTA, 25 mM Sodium fluoride, 0.1 mM sodium orthovandate, 0.1 mM PMSF, 0.2% Triton-X100, 0.3% NP-40, 100 nM Okadaic acid, and complete EDTA-free protease inhibitor) the beads were denatured at 99°C in 2× SDS–PAGE buffer and analyzed by SDS–PAGE and immunoblotting.

For GST pull down assays, 4 mg of cell lysate from the Nocodazole (100 nM) arrested mitotic HeLa cells was precleared by incubating with 25 $\mu$g bacterially expressed purified recombinant GST and 50 $\mu$l of 50% slurry glutathione sepharose beads (17-0756-01; GE Healthcare) for 2 h at 4°C. 2–4 mg of the precleared lysate was incubated with 20 $\mu$g of bacterially expressed purified recombinant GST or GST-Plk1PBD and 50 $\mu$l of 50% glutathione sepharose beads slurry for 4 h. After extensive washing in wash buffer (50 mM Tris, pH-7.4, 2 mM EDTA, 2 mM EGTA, 25 mM Sodium fluoride, 0.1 mM sodium orthovandate, 0.1 mM PMSF, 0.2% Triton-X100, 0.3% NP-40, 100 nM Okadaic acid, and complete EDTA-free protease inhibitor), the beads were denatured at 99°C in 2× SDS–PAGE buffer and analyzed by SDS–PAGE and immunoblotting.

For immunoblotting, HeLa cells synchronized with 100 nM Nocadazole for 16–20 h were lysed in lysis buffer (50 mM Tris, pH-7.4, 2 mM EDTA, 2 mM EGTA, 25 mM Sodium fluoride, 0.1 mM sodium orthovandate, 0.1 mM PMSF, 0.2% Triton-X100, 0.3% NP-40, 100 nM Okadaic acid, and complete EDTA-free protease inhibitor) for 2 h on ice. Cell lysate was denatured at denatured at 99°C in 2× SDS–PAGE buffer and analyzed by SDS–PAGE followed by immunoblotting. For immunoblotting, 1:1,000 rabbit anti-NuMA (sc-48773; Santa Cruz), 1:5,000 rabbit anti-GST (G-7781; Sigma-Aldrich), 1:500 mouse anti-Plk1 (sc-17783; Santa Cruz), and 1:5,000 rabbit anti-GFP (sc-8334; Santa Cruz) were used.

### Far-Western

Protein extracts from the Nocodazole arrested mitotic HeLa cells were resolved on SDS–PAGE and were transferred onto the PVDF membrane. Proteins on the membrane were denatured in Tris-based buffer (20 mM Tris, pH-7.5, 100 mM NaCl, 0.5 mM EDTA, 10% glycerol, 0.1% Tween-20, 2% skim milk powder, and 1 mM DTT) that also contains 6 M guanidine-HCl for 30 min at room temperature. This step is followed by gradual renaturation of the membrane-bound proteins in the same buffer but with 3 M and 1 M guanidine-HCl. The last renaturation step was performed with 0.1 M guanidine-HCl at 4°C. Subsequently, the blot was incubated overnight in the above-mentioned buffer without guanidine-HCl. The next day, the membrane was blocked with 5% skim milk powder in 1× PBST (containing 0.05% Tween 20) for 1 h at room temperature. Thereafter, the membrane was incubated with 5 $\mu$g (1 $\mu$g/$\mu$l) of recombinant GST/GST-PBD in protein binding buffer (20 mM Tris, pH-7.5, 100 mM NaCl, 0.5 mM EDTA, 10% glycerol, 0.1% Tween-20, 2% skim milk powder, and 1 mM DTT) at 4°C overnight. The membrane was washed with 1× PBST three times, 10 min each. The blot was then incubated with primary antibody against GST-fusion protein subsequent immunoblotting.

### In vitro kinase assay

To assay Plk1 kinase activity on recombinantly expressed C-terminus fragment of NuMA1700-2115, 1 $\mu$g of NuMA1700-2115 was incubated with 300 ng Plk1 kinase (14-777M; Merck) in kinase buffer (20 mM Hepes, pH-7.8, 15 mM KCl, 10 mM MgCl2, 1 mM EGTA, and 100 $\mu$g/ml BSA) supplemented with ATP and γ32P-ATP for 30 min at room temperature; the samples were analyzed by SDS–PAGE followed by autoradiography.

### Statistical analysis

To calculate the significance of the differences between the mean values obtained for the two different experiments two-tailed $t$ test was used as mentioned in the figure legends. $P$-value was considered to be significant if $P \leq 0.05$.

## Supplementary Information

## Acknowledgements

We thank Andreas Merdes, Daniel Gerlich, Arnaud Echard, Sivaram V S Mylavarapu, and Pierre Gönczy for providing us plasmids and cell lines. We are thankful to Iain Hagan, Sveta Chakrabarti, Cayetano González, Phong Tran, and Claudio Sunken for providing us critical comments on the manuscript. We thank Coralie Busso for her great help at the time of Kotak's lab set-up. We are grateful to the Department of Science and Technology (DST)-Fund for Improvement of Science and Technology infrastructure in universities & higher educational institutions (FIST), University Grants Commission Centre for the Advanced Study, Department of Biotechnology (DBT)-Indian Institute of Science (IISc) Partnership Program and IISc for the infrastructure support. This work is supported by the grants from the Wellcome Trust-DBT India Alliance (grant no. IA/I/15/2/502077) to S Kotak. S Kotak is a Wellcome Trust-DBT India Alliance Intermediate Fellow.

## Author Contributions

S Sana: data curation, formal analysis, investigation, and methodology.
R Keshri: data curation, formal analysis, investigation, and methodology.
A Rajeevan: data curation, formal analysis, investigation, and methodology.
S Kapoor: data curation.
S Kotak: conceptualization, resources, data curation, formal analysis, supervision, funding acquisition, investigation, methodology, and writing—original draft, project administration, review, and editing.

## Conflict of Interest Statement

The authors declare that they have no conflicts of interest.

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
