## [Reviewer comments · Life Science Alliance]

Life Science Alliance

Plk1 regulates spindle orientation by phosphorylating NuMA in human cells

Shrividya Sana, Riya Keshri, Ashwathi Rajeevan, Sukriti Kapoor, and Sachin Kotak
DOI: 10.26508/lsa.201800223

Corresponding author(s): Sachin Kotak, Indian Institute of Science (IISc)

Review Timeline:	Submission Date:	2018-10-23
	Editorial Decision:	2018-10-29
	Revision Received:	2018-11-05
	Accepted:	2018-11-08

Scientific Editor: Andrea Leibfried

Transaction Report:

Please note that the manuscript was previously reviewed at another journal and the reports were taken into account in the decision-making process at Life Science Alliance. Since the original reviews are not subject to Life Science Alliance's transparent review process policy, the reports and author response cannot be published.

October 29, 2018

RE: Life Science Alliance Manuscript #LSA-2018-00223-T

Dr. Sachin Kotak
Indian Institute of Science (IISc)
Microbiology and Cell Biology (MCB)
CV Raman Avenue
Bangalore, Karnataka 560012
India

Dear Dr. Kotak,

Thank you for transferring your revised manuscript entitled "Plk1 regulates spindle orientation by phosphorylating NuMA in human cells" to Life Science Alliance. Your manuscript was reviewed at another journal before twice, and the editors transferred those reports to us with your permission.

The reviewers who assessed your work elsewhere before appreciate your work, but would have expected further reaching mechanistic insight. This is not a concern for publication here. We would thus be happy to publish your work, pending that you address the remaining concerns the reviewers noted upon re-review at the other journal. To do so please provide a point-by-point response and slightly amend your manuscript text and data representation accordingly, including performing the requested statistical analyses. Please also see below for the formatting requirements we have (individual figure files needed and manuscript text as docx file, please).

A. FINAL FILES:

-- High-resolution figure, supplementary figure and video files uploaded as individual files: See our detailed guidelines for preparing your production-ready images, <http://life-science-alliance.org/authorguide>

-- Summary blurb (enter in submission system): A short text summarizing in a single sentence the study (max. 200 characters including spaces). This text is used in conjunction with the titles of papers, hence should be informative and complementary to the title. It should describe the context and significance of the findings for a general readership; it should be written in the present tense

and refer to the work in the third person. Author names should not be mentioned.

B. MANUSCRIPT ORGANIZATION AND FORMATTING:

Full guidelines are available on our Instructions for Authors page, <http://life-science-alliance.org/authorguide>

Thank you for your attention to these final processing requirements.

Sincerely,

November 8, 2018

RE: Life Science Alliance Manuscript #LSA-2018-00223-TR

Dr. Sachin Kotak
Indian Institute of Science (IISc)
Microbiology and Cell Biology (MCB)
CV Raman Avenue
Bangalore, Karnataka 560012
India

Dear Dr. Kotak,

Thank you for submitting your revised Research Article entitled "Plk1 regulates spindle orientation by phosphorylating NuMA in human cells" to Life Science Alliance. We appreciate the introduced changes and it is a pleasure to let you know that your manuscript is now accepted for publication in Life Science Alliance. Congratulations on this interesting work.

DISTRIBUTION OF MATERIALS:

Again, congratulations on a very nice paper. I hope you found the review process to be constructive and are pleased with how the manuscript was handled editorially. We look forward to future exciting submissions from your lab.

Sincerely,

Andrea Leibfried, PhD
Executive Editor

Life Science Alliance
Meyerhofstr. 1
69117 Heidelberg, Germany
t +49 6221 8891 502
e a.leibfried@life-science-alliance.org
www.life-science-alliance.org